# Gum Arabic Nanoparticles as Green Corrosion Inhibitor for Reinforced Concrete Exposed to Carbon Dioxide Environment

**DOI:** 10.3390/ma14247867

**Published:** 2021-12-19

**Authors:** Mohammad Ali Asaad, Ghasan Fahim Huseien, Mohammad Hajmohammadian Baghban, Pandian Bothi Raja, Roman Fediuk, Iman Faridmehr, Fahed Alrshoudi

**Affiliations:** 1Department of Civil Engineering, Iraq University College, IUC Al-Estiqlal St., Basra 61007, Iraq; mohammad.asaad@iuc.edu.iq; 2Department of the Build Environment, School of Design and Environment, National University of Singapore, Singapore 117566, Singapore; 3Department of Manufacturing and Civil Engineering, Faculty of Engineering, Norwegian University of Science and Technology (NTNU), 2815 Gjøvik, Norway; 4School of Chemical Sciences, Universiti Sains Malaysia, Penang 11800, Malaysia; bothiraja@usm.my; 5School of Engineering, Far Eastern Federal University, 690091 Vladivostok, Russia; roman44@yandex.ru; 6Institute of Architecture and Construction, South Ural State University, Lenin Prospect 76, 454080 Chelyabinsk, Russia; s.k.k-co@live.com; 7Department of Civil Engineering, College of Engineering, King Saud University, Riyadh 11421, Saudi Arabia

**Keywords:** GA-NPs, green corrosion inhibitor, carbonation resistance, rebar corrosion, depth carbonation, morphology

## Abstract

The inhibiting effect of Gum Arabic-nanoparticles (GA-NPs) to control the corrosion of reinforced concrete that exposed to carbon dioxide environment for 180 days has been investigated. The steel reinforcement of concrete in presence and absence of GA-NPs were examined using various standard techniques. The physical/surface changes of steel reinforcement was screened using weight loss measurement, electrochemical impedance spectroscopy (EIS), atomic force microscopy and scanning electron microscopy (SEM). In addition, the carbonation resistance of concrete as well screened using visual inspection (carbonation depth), concrete alkalinity (pH), thermogravimetric analysis (TGA), SEM, energy-dispersive X-ray spectroscopy (EDX) and X-ray diffraction (XRD). The GA-NPs inhibitor size was also confirmed by transmission electron microscopy (TEM). The results obtained revealed that incorporation of 3% GA-NPs inhibitor into concrete inhibited the corrosion process via adsorption of inhibitor molecules over the steel reinforcement surface resulting of a protective layer formation. Thus, the inhibition efficiency was found to increase up-to 94.5% with decreasing corrosion rate up-to 0.57 × 10^−3^ mm/year. Besides, the results also make evident the presence of GA-NPs inhibitor, ascribed to the consumption of calcium hydroxide, and reduced the Ca/Si to 3.72% and 0.69% respectively. Hence, C-S-H gel was developed and pH was increased by 9.27% and 12.5, respectively. It can be concluded that green GA-NPs have significant corrosion inhibition potential and improve the carbonation resistance of the concrete matrix to acquire durable reinforced concrete structures.

## 1. Introduction

The design of concrete structure must have durability, safety, aesthetics and serviceability for entire life duration. Consequently, the most significant aspects of a construction material success are mechanical and durability of the concrete’s performance [1]. In the global construction industry, the major apprehension about reinforced concrete structure is its early deterioration as a result of corrosion of the reinforcement [2,3]. The existence of corrosion in concrete structures is due to the steel surface’s depassivation, and this occurs throughout the penetration of reinforced concrete by one or both of the factors; chloride ions (Cl^−^) or carbon dioxide (CO_2_), and such issues result in major costs in concrete structures maintenance globally [4,5]. 

Amongst the most corrosive factors, concrete carbonation is one of the significant reasons that have a negative impact on the concrete’s durability. This process takes place when the CO_2_ gas dissolved in water or from the atmosphere is reacting with free hydroxides, which are mainly made up of Ca(OH)_2_ (calcium hydroxide) in concrete, to develop calcium carbonate [5,6]. Besides, CO_2_ gas also reacts with calcium–silicate–hydroxide gel in the concrete matrix thus further forming calcium carbonate [7]. Since the majority of concrete structures are in contact with the atmosphere, corrosion due to carbonation is a significant worry. In brief, the carbonation process occurs by the following stages: (i) diffusion of CO_2_ into the concrete, (ii) reaction of the CO_2_ with Ca(OH)_2_ (calcium hydroxide) in the presence of moisture, (iii) a reduction in pH from around 12.5 to 8.0 and (iv) de-passivation of the steel at this lowered pH.

To control such defects, varied prevention strategies have been proposed, all aimed to stop, delay or slow down the corrosion mechanism including cathodic protection systems [8], stainless steel bars [9], treatment of concrete surface using low permeability concrete [10] and galvanized steel bars [11]; and is also used in silica fumes [12], fibre glass [13], epoxy-coatings [14,15], super-hydrophobic anti-corrosion coating [16], adding lauric acid into concrete [17] and corrosion inhibitors [18,19].

One of the renowned ways of controlling and reducing the corrosion rate of the low carbon steel is to apply organic inhibitors [15,20,21,22,23,24,25,26,27]. These organic corrosion inhibitors are either simple or mixtures that are added to the aggressive environments in low quantities in order to reduce, control or even hinder reactions from occurring between the metal and its surroundings [28]. The effectiveness of organic corrosion inhibitors is due to the presence of N, O or S atoms that are the core of the formation of adsorption process; of which that prevents the active sites of metals from corrosive media exposure thus decreasing the corrosion rate [29,30,31,32]. However, avoiding the use of commercial corrosion inhibitors to protect the mild steel in harsh environments is related to their being hazardous to the environment and highly toxic [31,33,34,35]. Consequently, due to safety concerns, the researchers have focused extensively on developing effective organic inhibitors from natural ingredients, such as extracts from fruits, plants and peels, that are eco-friendly and harmless, which are also known as green corrosion inhibitors. Several studies have been conducted and published on the application of natural products as corrosion inhibitors on mild steel in different harsh environments such as, *Artemisia pallens* [34], *Neolamarckia cadamba* [36], *Rhizophora apiculate* [37], *Musa paradisiac* [38], aloe vera [29], apricot juice [39], *Juglans regia* [40], Asafoetida [41] and Pomelo [42]. Results have shown that the organic green corrosion inhibitors have an inhibition efficiency of 65–97%. The inhibitive impact of natural compounds is ascribed to the ability of green inhibitor molecules to adsorb over metal surfaces thus, formation of a thin preventive layer and blocking the active sites.

Pertaining to the application of green inhibitor upon the reinforced concrete (RC), Abdulrahman and Ismail [43,44] have studied the effects whereby 2–4% of the green inhibitor known as *Bambusa Arundinacea* is applied onto contaminated concrete by sulfate and chloride. Loto et al. [45] have also studied the effects of another green inhibitor, *Vernonia amygdalina* (bitter leaf extract) with concentrations of 25–100% being applied onto steel reinforcement in concrete with exposure to 3.5% sodium chloride. The results have shown that the bitter leaf extract has fair corrosion inhibition concentrations of 50% and 75%. While the optimum inhibition efficiency of 90% was achieved when the inhibitor concentration examined is at its lowest at 25%. Meanwhile, the effect of another corrosion inhibitor, *Vernonia amygdalina* was studied by Eyu et al. [46] along with sodium and calcium nitrate onto steel-reinforced concrete exposed to 3.5% NaCl solution for 70 days. The authors noted that the *Vernonia amygdalina* inhibitor is more effective compared to calcium or sodium nitrate in terms of reduction of corrosion rate for steel within concrete for the duration of the immersion. Another study by Okeniyi et al. [47] was conducted where admixtures of different concentration levels of *Anthocleista djalonensis* leaf extract were incorporated into steel reinforced concrete exposed to the saline medium. They detected that the maximum inhibition efficiency of 97.43% was achieved at 0.4167% of a green inhibitor concentration. Our research team successfully reported the incorporation of 5% *Elaeis guineensis*/AgNPs into the reinforced concrete that found resulting in improving inhibition efficiency up to 95% [48]. This is due to the formation of additional C-S-H gel, which is responsible for blocking the pores within the concrete matrix. Gular et al. [49] investigated the impact of incorporation of different percentages (0.5, 1 and 1.5%) of nanomaterials in the concrete matrix such as nano-Fe_2_O_3_, nano-TiO_2_, nano-Al_2_O_3_ and nano-SiO_2_ on mechanical properties. They concluded that the presence of 1.5% of nano-Al_2_O_3_, and nano-SiO_2_ indicated enhanced mechanical properties of concrete by up to 22% at 28 days in comparison with other nanoparticles. Moreover, several [50,51,52,53,54,55,56] authors have been studying the effect of nanoparticles on mechanical properties and corrosion of reinforced concrete with a variety of percentages (1–5%), and they found that adding the nanoparticles of concrete can be increased the mechanical properties and the concrete durability at different ages.

In continuation of our earlier investigation, the present study reports Arabic gum-nanoparticles as corrosion inhibitors for reinforced steel in concrete that exposed over carbonated environment. Gum Arabic (GA) investigated was extracted in the form of exudate from the stem and branches of the trees, *Acacia senegal* tree [57,58]. The GA consists of mixture of biopolymers, which includes amphiphilic polysaccharide-protein complexes that leads to the stabilisation and formation of emulsions [59,60]. As a hydrocolloid, GA has low-viscosity when its concentration is high with exceptional water solubility in comparison with other gums [61,62]. GA was one of the earliest biopolymers being applied in both food and non-food products especially within the cosmetics and medicine industries as thickening, stabiliser and emulsifier agents due to its beneficial properties such as pH stability, non-toxic, renewability, biocompatibility, gelling, low cost and high solubility [63,64]. Additional application of GA includes the synthesis and modification of numerous metallic nanoparticles (metal oxides, gold and silver nanoparticles) [65]. Further, GA has been reported as a green corrosion inhibitor that displays inhibition efficiency of 97% for mild steel being exposed to acidic substances [66,67,68,69]. However, to the best knowledge to date, there has been no study of the inhibition efficiency on the GA upon reinforced concrete. Hence, the present study is attempted to analyse the corrosion inhibition potential of GA-nanoparticles on reinforced steel in concrete structures that are exposed to carbon dioxide environment. Standard techniques like weight loss, electrochemical impedance spectroscopy, pH and carbonation depth tests were carried out to study the mechanism of corrosion inhibition for reinforced concrete specimens exposed to CO_2_ environment. Further, morphology of steel reinforcement surfaces was screened via SEM and AFM and also the morphology of concrete specimens was examined by SEM, EDX, XRD and TGA. Finally, the powder of GA-NPs was characterized via transmission electron microscopy (TEM) to detect the particle size.

## 2. Materials and Methods

### 2.1. Green GA-NPs Inhibitor Preparation

The dried Gum Arabic (GA) specimens were procured from *Acacia Senegal* trees exuded was locally available and directly purchased from Alsaadi Company for Aromatics and spices, Basra, Iraq, was kindly provided by Dr. Mohammad Ali Asaad (Iraq University College, Basra, Iraq), and ground into powder. In order to obtain the fine size of extracted (nanoparticles), 1000 g of resultant powder was dissolved in 4000 mL distilled water for 24 h at ambient temperature (28 ± 2 °C). Afterward, the suspension was stirred for 3 h at 45 °C and then filtered using filtration paper (Whatman) grade 1 (Whatman, Taufkirchen, Germany). Lastly, the resulting mixture was centrifuged (Hettich, EBA 21 Model, Tokyo, Japan) at 4500× g rpm for 30 min to achieve the GA-nanoparticles inhibitor. The characteristics of *Acacia Senegal* (gum Arabic) are listed in Table 1 [70]. According to Ali et al. [71], the arabinogalactan is the most component of the GA with 88.4% in total. The arabinogalactan possesses a low molecular mass and low protein content of 3.8 × 10^5^ (g/mol) and 0.35% respectively. The molecular structure of arabinogalactan is depicted in Figure 1 [67].

### 2.2. Transmission Electron Microscope (TEM)

The morphology (shape and size) of the GA-NPs powder was determined using BIO-TEM, model Hitachi-HT 7700, Tokyo, Japan. Distilled water was used to disperse the nanoparticles of the green GA inhibitor under ultrasonic treatment, and then a drop of the solution was placed onto the carbon-coated copper grids (Hitachi-HT 7700, Tokyo, Japan) and was investigated at 120 kV accelerated voltage.

### 2.3. Materials and Concrete Specimens Preparation

First phase, OPC—ordinary Portland cement (type I) (Falcon Cement Company, Hafirah, Bahrain) was prepared in accordance with ASTM C 150 [72] as a concrete component and used for all mixes design. The physical property and chemical composition of such OPC were summarised in our earlier studied [48]. River sand (Al-faw, Basra, Iraq) having a specific gravity of 2.55, a density of 1630 kg/m^3^ and fineness modulus of 2.57 was sieved by sieve (W.S. Tyler, Mentor, OH, USA) number of 4.57 mm and used as a fine aggregate. The crushed stone (Al-faw, Basra, Iraq) had a minimum and maximum particle size as well as bulk density of 5 mm, 9.5 mm and 2700 kg/m^3^ respectively, was used as coarse aggregate. Normal fresh water in a w/c ratio of 0.55 and water content of 217 kg/m^3^ was used in all concrete mixtures.

Second phase, a concrete slab with dimensions of 200 mm (length), 180 mm (width) and 66 mm (thickness) was designed for corrosion examinations. Steel reinforcement bars (Ransheng Steel, Tianjin, China) specimens of 225 mm in length and 16 mm diameter acting as a working electrode were cut by metal cutting machine (Jiangsu Goldmoon Industry Co., Jiangsu, China) and then polished using different grades (600, 800, 1000, 1200 and 1600) of emery papers (Jiangsu Goldmoon Industry Co., Jiangsu, China). Next, the steel bars specimens were degreased with concentrated acetone (70%) (Qrëc, Chonburi, Thailand), washed in distilled water, air-dried and then embedded in the middle of the concrete slab, thus a 25 mm of concrete cover was provided around the steel bar on both sides and from the base, in order to provide limited corrosion prevention for steel reinforcement.

In the third phase, based on the mass of cement, GA-NPs (3%) were mixed with water and added into concrete mix components during the mixing process and then cast into desired moulds (Ransheng steel, Tianjin, China). The mix proportion of concrete specimens was indicated in Table 2. In order to protect the steel bars from crevice corrosion after concrete casting, the protrusion of steel bar (5 cm—the rest of working electrode) was isolated with silicone sealant (McCoy Soudal, Delhi, India). Furthermore, concrete cube specimens of dimension 100 mm^3^ were also cast, which were further used to investigate the influence of green GA-NPs inhibitor on accelerated carbonation and carbonation depth. Finally, after one day of casting, all concrete specimens were demoulded, and then cured in fresh water at 25 °C and relative humidity of 75% ± 5% for 28-day prior to move to CO_2_ gas chamber (MR, Sharjah, U.A.E). The compressive strength of concrete was designed for 30 MPa at 28 days using 1:1.73:2.8 concrete mix design.

### 2.4. Electrochemical Impedance Spectroscopy Test

Electrochemical impedance spectroscopy (EIS) (Ametek Scientific Instruments, Seattle, WA, USA) was conducted in a three electrode cell comprising of saturated calomel (SCE), steel specimens and platinum wire as reference, working and counter electrodes respectively, while NaCl of 3.5% (Green Research scientific, Basra, Iraq) was used as an electrolyte for EIS experiments. Every measurement was examined using an electrochemical workstation model VersaStat 3 (Princeton, Singapore) at 25 ± 2 °C, following the E_OCP_—open-circuit potential was stabilised for 30 min over perturbation of 10 mV (AC sine wave, peak-to-peak), a frequency range of 1000–100 Hz. Next, ZSimpWin software 3.2 was utilised for fitting impedance data with several sets of equivalent circuit. Furthermore, the electrochemical impedance spectroscopy parameters were calculated by inserting the following data: the density and the equivalent weight of rebar specimens is 7.85 (g·cm^−3^), and 27.92 (g) respectively, while the area of exposure of steel reinforcement is 68.36 (cm^2^). In addition, the EIS test were carried out for steel reinforcing bars in concrete that subjected to CO_2_ gas environment for 28, 90 and 180-day in the absence and presence of 3% GA-NPs inhibitor. The charge transfer resistance (R_s_) and double-layer capacitance (C_dl_) data were obtained from the diameter of the semicircles of the Nyquist plot. Based on the charge transfer resistance (R_ct_) data, the inhibition efficiency (IE%) of reinforced concrete was calculated from the following formula [73]:(1)$$\mathrm{IE}\%=\left(1-\frac{R_{\mathrm{ct}}^{o}}{R_{\mathrm{ct}}^{i}}\right)\times 100$$
where $R_{\mathrm{ct}}^{i}$ and $R_{\mathrm{ct}}^{o}$ denote the charge transfer resistance of reinforcing steel with and without GA-NPs inhibitor respectively.

### 2.5. Gravimetric Measurements for Concrete Slab

Gravimetric or weight loss measurements was carried out to determine the corrosion rate and inhibitor efficiency of concrete slabs exposed to CO_2_ gas at 28, 90 and 180 days. The slabs were broken using the splitting tensile machine (3000 kN capacity NL Scientific, Selangor, Malaysia). Afterward, the steel specimens extracted from the slabs and cleaned using cleaning solution in accordance with ASTM G1 [74], which is consisted of 500 mL of 37% (concentrated grade) HCl acid (Quality Research Chemicals, Selangor, Malaysia) with 3.5 g of hexamethylenetetramine (Sigma-Aldrich, Taufkirchen, Germany) added with distilled water to make the volume up to 1000 mL. Each specimen was cleaned several times, after which the weight loss was evaluated by calculating the difference between the initial and final weights of the specimens. From the results, the inhibitor efficiency (IE%), corrosion rate (CR, mm/year) and surface coverage (θ) were obtained using the following equations respectively [32]:(2)$$\mathrm{IE}\%=\left(1-\frac{W_{i}}{W_{o}}\right)\times 100$$
where W_i_ and W_o_ are the weight loss values of reinforcement bar in the presence and in the absence and GA-NPs.
(3)$$\mathrm{CR}=\frac{87,600\;W}{\rho \times t\times A}$$
where W is the weight loss of the rebar (g), ρ is the rebar density (7.85 g·cm^−3^), t is the time of exposure (h) and A is the rebar surface area (cm^2^).
(4)θ=(IE%)/100

The term “surface coverage (θ)” refers to the “inhibitory zone/area over the metal surface” covered by the inhibitor via the adsorption process.

### 2.6. Carbonation Depth Test

After 28-day of curing the specimens in normal water, all the concrete cube specimens were removed and placed in the chamber. In line with the recommendations outlined by Sawada et al. [75], concrete specimens were subjected to accelerated carbonation by creating an atmosphere of 65–75 relative humidity (RH) in enclosed plastic tanks (MR, Sharjah, U.A.E), passing through carbon dioxide (CO_2_) gas (Almansour, Basra, Iraq) for a period of 30 min, conducting this two times each day at a temperature of 25–30 °C. A pressure gauge was affixed to the carbon dioxide gas cylinder to observe the pressure inside the chamber. Next, the chamber was tightly closed and vacuumed for 2 min under a pressure of around 600 mmHg. Then, CO_2_ gas was allowed to pass to the chamber at a pressure of 750 Psi. A regulator (Tianjin Sure Instrument Co., Tianjin, China) was affixed to the CO_2_ gas cylinder to control the pressure of CO_2_ inside the chamber. The specimens were subjected to this treatment for 28, 90 and 180 days. Once the carbon dioxide exposure had been completed, three specimens of untreated and treated concrete with GA inhibitor were subjected to splitting at 28, 90 and 180 days and their centres were sprayed with phenolphthalein solution to evaluate carbonation. Here, carbonation depth was identified as being the length of space between the coloured area’s edge and the outer surface of the concrete, the indicator of carbonation depth serves as a useful marker of the degradation that the specimens have incurred from carbonation attacks [76]. Whereas normal concrete treated with 1% of phenolphthalein solution will turn pinkish purple in colour, concrete that has undergone carbonation will show no colour change [77]. A 1% solution of phenolphthalein (Sigma-Aldrich, Taufkirchen, Germany) was prepared by dissolving 1 mg of phenolphthalein indicator powder in 90 cc of 2-propanol (Iso-propanol, Qrëc, Chonburi, Thailand) and distilled water was added to make the solution volume up to 100 cc. The concrete specimens were split into two parts and instantly sprayed via phenolphthalein solution. The depth of the uncoloured (carbonated) layer below the external surface was measured to the nearest mm at four locations, and the mean value was recorded.

### 2.7. PH Measurement

The alkalinity variation of carbonated concrete specimens was evaluated by calibrating the pH value of concrete powder at depths of 2–20 mm at 180-day of exposure to CO_2_. The concrete specimens were drilled from the external to the internal surfaces, and 1 g of powder at each different depth was collected. Next, the collected powder was added to 50 mL of distilled water as a solvent and the entire mixture was stirred for 24 h at 25 °C. A pH meter (Mettler-Toledo AG, Columbus, OH, USA) was used to calibrate the pH value of specimens exposed to carbon dioxide at different depths.

### 2.8. Morphological Analysis of Rebar Surface

The surface morphology of steel reinforcement specimens in the absence and presence of 3% GA-NPs was carried out using SEM-scanning electron microscope model Jeol, JSM-IT300, Tokyo, Japan, and atomic force microscope (AFM) model Nano-Wizard 3, Tokyo, Japan. Following 180 days of exposure to CO_2_, the concrete slabs were split and the embedded carbon steel bars were carefully removed, rinsed in distilled water, air-dried, cut into small pieces with 16 mm in diameter and 10 mm thickness, and then used for morphologies scanned.

### 2.9. Morphological Analysis of Concrete

#### 2.9.1. Thermal Gravimetric Analysis (TGA)

Following 180 days of exposure to CO_2_ gas, concrete specimens were broken and small pieces of dimensions 10 × 10 × 7 mm were extracted from the core of specimens and then ground into powder. The powder specimens in the presence and absence of GA-NPs inhibitor were subjected to TGA/DTA thermograms analyser model TGA–Q 500, Cincinnati, OH, USA in order to detect the percentage of weight loss during the thermal degradation. Next, concrete powder of 2 g was placed in a ceramic pan having a height and diameter of 5 and 6 mm, respectively, and then subjected to heating at a temperature of 30–1000 °C. The decomposition of Ca(OH)_2_ concentrates was observed at temperature range of 400 to 500 °C and the C-S-H dehydration was also resulted in the weight loss at temperature range of 600–700 °C [78,79]. In addition, the percentages of the presence of both C-S-H gel (calcium silicate hydrate) due to the dehydroxylation of Ca(OH)_2_ and calcium hydroxide Ca(OH)_2_ content in the concrete matrix were determined according to the following formula [80]:(5)$$C-S-H\left(\%\right)=\mathrm{Total}\;\mathrm{LOI}-{\mathrm{LOI}}_{\mathrm{CC}}-{\mathrm{LOI}}_{\mathrm{CH}}$$
whereas LOI_CH_ represents the dehydration of Ca(OH)_2_ at a temperature of 400–550 °C, and LOI_CC_ represents the loss of CO_2_ at a temperature of 600–750 °C range. According to Singh et al. [81] the amount of CH (calcium hydroxide) can be determined precisely from the TGA curve according to the formula:(6)$$\mathrm{CH}\left(\%\right)={\mathrm{WL}}_{\mathrm{CH}}\left(\%\right)\times \frac{{\mathrm{MW}}_{\mathrm{CH}}}{{\mathrm{MW}}_{H}}$$
whereas MW_H_ and MW_CH_ represent the molecular weights of water (18 g/mol) and CH (74 g/mol), respectively, while WL_CH_ represents the weight loss of CH dehydration.

#### 2.9.2. SEM and XRD Analysis for Concrete

SEM (Jeol, JSM-IT300, Tokyo, Japan) equipped with EDX-energy dispersive spectroscopy, was utilised to examine the morphology of concrete specimens with and without GA-NPs inhibitor after 180-day of exposure to CO_2_. Small pieces of crushed carbonated concrete having a dimension of 14 mm × 14 mm × 5 mm were collected from the core of concrete cubes after subject them to split. Then, the specimens were transferred to vacuum environment up to 50 °C, till the constant mass of specimens was observed. Finally, the specimens were placed on cylinder stub and subjected to an automated platinum sputter coater (Model-Quorum (Q150R), Henan, China) for 1.5 min prior to testing.

The XRD pattern for concrete specimens treated and untreated with GA-NPs inhibitor was measured using model Rigaku, SmartLab 3 kW, Tokyo, Japan. The specimens were collected and ground into powder using a grinding machine (Panasonic, Osaka, Japan). The powder was located on the sample holder, run at (30 mA/40 kV), scanned at 2-theta angle from 20–80° by scanning rate of 5°/min, and X-rays of (k = 1.5406 Å) created by a Cu K_α_ source.

## 3. Results and Discussion

### 3.1. Transmission Electron Microscope (TEM)

Figure 2 illustrates the TEM micrograph of GA-NPs inhibitor, whereas the GA-nanoparticles were distributed in a non-agglomerated with spherical shapes, and having a scale ranging from 9.27–123.69 nm. However, the distributions of nanoparticles were fitted using a Gaussian fit curve as shown in Figure 3 in order to determine the average size of all particles. The peak value of GA-NPs distribution size was found at 40.24 nm. This size can be considered as a nano-scale according to Ye et al. [82] which is defined the nanomaterials as materials consisting of particles less than 100 nm in size.

### 3.2. Weight Loss

Table 3 shows the results of non-electrochemical measurements which are included weight loss, corrosion rate, the inhibitor efficiency and coverage area for the concrete specimens following 28, 90 and 180 days of exposure to carbon dioxide gas. The contact between concrete structure and atmosphere allowed the carbonation process to initiate according to the following equations [7]:(7)$${\mathrm{CO}}_{2}+H_{2}O\to H_{2}{\mathrm{CO}}_{3}$$
(8)$$\mathrm{Ca}{\left(\mathrm{OH}\right)}_{2}+{\mathrm{CO}}_{2}\to {\mathrm{CaCO}}_{3}+H_{2}O$$

As a consequence of these reactions, the alkalinity (pH) of the concrete is lowered, which is led to producing voluminous rust over steel reinforcement. This can impact the reduction in the thickness of the steel reinforcement, and general degradation [83], hence, the steel reinforcement bar is losing its own weight. The maximum weight loss (0.6775 g) and corrosion rate (CR) (9.87 × 10^−3^ mm/year) were obtained for the control specimen. These results indicate that CO_2_ penetrated the concrete cover and reached the streel reinforcement surface. Hence, destroyed the passive film and initiated of corrosion process. By contrast, the minimum weight loss value (0.0393 g), minimum corrosion rate (0.57 × 10^−3^ mm/year), and maximum end-of-test inhibitor efficiency (94.2%) were obtained for the specimen treated with 3% GA-NPs inhibitor. The observed increase in efficiency (%) with exposure time is attributed to the increasing surface coverage (θ) by adsorbed inhibitor molecules over the steel reinforcement bar to form a corrosion resistant layer during the exposure period [35]. Thus, the formation of a resistant layer over the steel reinforcement led to the isolation of the corrosive agents to reach the steel surface and react with the iron, which inhibited the corrosion process.

### 3.3. Electrochemical Impedance Spectroscopy

Nyquist plots (Figure 4) represent the EIS results of reinforced steel in concrete with the presence and absence of 3% GA-NPs inhibitor that were obtained after 28, 90, and 180 days of exposure to CO_2_. According to literature [38,84,85], the small high-frequency semicircle obtained both in the absence and presence of GA-NPs constitute a characteristic response of the time constant of the double-layer capacitance (C_dl_) and charge-transfer resistance (R_ct_). However, the semi-circle loops that appeared in the high-frequency region demonstrated the function of the charge transfer process in controlling the electrode reaction, while the presence of a large depressed semicircle (depressed capacitive loops) of modified steel reinforcement with GA-NPs that exposed to CO_2_ for 180 days extending from the high to low-frequency regions, indicating that the primary effect of the charge-transfer resistance on the corrosion reaction results from the development of a protective film due to the adsorption of inhibitor molecules onto the steel surface. Besides, the depressed nature of the semicircles is usually related to the roughness and inhomogeneity of the solid surface.

Figure 5 depicts the equivalent Randles circuit which is utilised to fit the impedance results of Nyquist plots for the rebars embedded into concrete with and without the inhibitors, in which resistor (R_s_) represents the ohmic resistance of the system, and resistor (R_ct_) represents the resistance of the inhibitor to the charge transfer process during metal oxidation, whereas (Q) is introduced as the double layer capacitance (C_dl_). In fact, various equivalent circuits were generated and fitted against the experimental impedance values, while the circuit that having lowest fitting error was selected and reported here.

The precise examination of the impedance parameters at 28, 90 and 180 days are presented in Table 4. From the results, it is evident that concrete specimens treated with 3% GA-NPs found reduced the C_dl_-double layer capacitance and enhances the R_ct_-charge transfer resistance, so that a great diameter semicircle is detected in the Nyquist plots, while the ability of GA-NPs corrosion inhibitor to adsorb upon the electrode surface may explain the observed decrease in C_dl_.

The observed increase in R_ct_ values with time from 20,400 to 49,600 Ω·cm^2^ (Table 4) following treatment of the concrete with 3% GA-NPs inhibitor may be due to the precipitation impact of the solid Ca(OH)_2_ layer which is led to the reduction of concrete permeability and the diffusion of carbon dioxide due to the development of C-(A)-S-H gel [86], and by the formation of a barrier film at the steel-concrete interface. The formation of a barrier film has been demonstrated by the increasing surface coverage values (θ) from 0.83 to 0.95 in the presence of the 3% GA-NPs inhibitor, hence the present results confirm that the GA-NPs extract hindered the corrosion agents to reach the embedded steel in concrete by means of a surface adsorption mechanism. The corrosion inhibition potential of Gum Arabic inhibitor is most likely related to its adsorption abilities over the reinforcement steel surface. In the present study, Gum Arabic inhibitor adsorbed on the steel surface and the interactions between the electron pairs of phytoconstituents and the reinforcement material that forming a protective layer, and preventing the reinforcement steel from the direct attack of the aggressive substances surrounding it [87]. Adsorption process can be described via following mode of interaction: (a) physisorption mode–electrostatic forces of interaction between Gum Arabic molecules and the electric charge over the steel reinforcement /solution interface; (b) chemisorption mode–electrons/sharing from Gum Arabic molecule to the reinforcement steel surface that resulted in covalent type bonding [88]. After the Gum Arabic molecules are adsorbed over the reinforced steel surface, that can shield the whole steel surface, when the amount of Gum Arabic molecules are sufficient enough to form a protective layer adsorption structure on the steel surface, immaterial that it may be an anodic/cathodic zone. Hence, the shielding effect of protective film, electrons/aggressive ion transferring from the electrolytes in and out of reinforcement structures is obstructed that resulted in efficient corrosion protection for reinforcement components [89].

Furthermore, the results presented in Table 4 indicate a higher inhibitor efficiency and lower double-layer capacitance over the whole exposure period in the presence of the GA-NPs. The maximum inhibition efficiency (94.5%) and minimum C_dl_ (0.0138 × 10^−5^ μF·cm^−2^) were obtained in the presence of GA-NPs inhibitor after 180 days of exposure. These results may be explained by the hydrophobic capacitive nature of γ-Fe_2_O_3_ film that established upon the surface of reinforcing steel, which gives a huge time constant indicating passivation of the steel. This can be compared with increased the C_dl_ of (7.68 × 10^−5^ μF·cm^−2^) for control specimens at the end of test due to diffusion of CO_2_ in concrete. The regions of continuously increasing real and imaginary impedance (Figure 4), have the effect of shielding the steel from corrosion, hence providing corrosion resistance. The observed impedance of the GA-NPs inhibitor is due to slow oxygen diffusion through the concrete matrix and to the solid hydroxide layer at the steel-concrete interface providing a dielectric film component [90,91,92].

Aguiar and Júnior [7] confirmed that 50% of CO_2_ penetration into concrete reacts with C-S-H while the other 50% reacts with calcium hydroxide which is mainly attributed to reduce the pH of concrete. Therefore, the Nyquist plot (Figure 4) for the control concrete specimen presents a shortened depressed semicircle. This indicates break-down of the rebar passive film due to reduce of concrete alkalinity by carbon dioxide diffusion, leading to the observed reduction in the charge transfer resistance.

### 3.4. Morphology of Rebar Surface

#### 3.4.1. SEM Analysis

SEM micrographs of steel reinforcement surface of concrete treated with and without 3% GA-NPs following 180-day of exposure to CO_2_ are depicted in Figure 6. From Figure 6a, it is evident that the surface of reinforcing steel without GA-NPs inhibitor exhibited rough, damaged and corroded surface, as well as the cracks were also formed. In contrast, the SEM morphology of reinforcing steel surface treated with green GA-NPs inhibitor at same magnification exhibited a smooth surface without any cracks which is indicated that the adsorption of inhibitor molecules. Hence, providing a prevention layer that hindered or reduced the diffusion of CO_2_ to react with Fe and de-passivate the steel reinforcement film. Carbon monoxide can be reacted with iron dioxide in many processes to produce carbon dioxide and iron as follows:(9)$$3{\mathrm{Fe}}_{2}O_{3}+\mathrm{CO}\to 2{\mathrm{Fe}}_{3}O_{4}+{\mathrm{CO}}_{2}$$
(10)$${\mathrm{Fe}}_{3}O_{4}+\mathrm{CO}\to 3\mathrm{FeO}+{\mathrm{CO}}_{2}$$
(11)$$\mathrm{FeO}+\mathrm{CO}\to \mathrm{Fe}+{\mathrm{CO}}_{2}$$

#### 3.4.2. Atomic Force Microscopy (AFM)

The 2D and 3D AFM morphologies of the steel reinforcement surface for untreated and treated concrete with GA-NPs inhibitor are depicted in Figure 7 respectively. From Figure 7a,b, it can be seen that the surface of steel reinforcement in the absence of GA-NPs inhibitor suffered from being severely corroded and highly damage with an average surface roughness of 2.186 μm. The high average surface roughness can be ascribed to the development of corrosion products, essentially, magnetite-Fe_3_O_4_ and iron (II) carbonate-FeCO_3_ due to the diffusion of CO_2_. Conversely, the image of steel reinforcement surface (Figure 7c,d) in the presence of GA-NPs inhibitor was significantly enhanced, so that the average surface roughness was reduced to 546.6 nm with an improvement of 75% with respect to control steel reinforcement (without inhibitor). This observation further evident that the adsorption of inhibitor and the formation of a thin film that hindered the penetration of CO_2_ to reach the steel reinforcement surface. These findings robustly support the outcomes obtained by weight loss and EIS experiments.

### 3.5. Visual Inspection on Accelerated Carbonation of Concrete

Figure 8 illustrates the visual inspections of accelerated carbonated specimens (including and not including GA-NPs inhibitor) after 180 days of exposure to CO_2_. It can be seen that the cube concrete specimen (cube) is associated with smaller carbonation depths when 3% GA-NPs inhibitor is present and when compared to control specimen. Figure 9 shown the carbonation depths (mm) for concrete specimens exposed to CO_2_ gas for 28, 90 and 180 days with and without GA-NPs inhibitor. The carbonation depths of control specimens were measured as 4.1 mm, 19.6 mm and 34.4 mm at 7, 28 and 180 days of exposure, respectively, while 3.0, 6.2 and 12.3 mm carbonation depths were recorded for modified concrete specimens with GA-NPs inhibitor at the same period of exposure. This implies that the GA-NPs inhibitor is linked to the clearest impact to enhance the concrete matrix which led to lowering carbonation depth by 64.42% following 180 days of exposure to CO_2_ gas with respect to control specimens. Moreover, the high carbonation depth of control specimens is not surprising. The reason is that control concrete is inclusive of numerous internal pores especially with a high w/c ratio (0.55), and this means that it can be subjected to carbonization is a clearer way. Kim et al. [93] observed that the increasing of w/c ratio of concrete causes increasing in concrete porosity, hence diffusion of carbon dioxide gas is greater in control concrete. Nevertheless, it should be noted that the incorporation of tiny size (40.24 nm) of GA inhibitory to concrete can act as a nano-filler reduced and/or blocked the internal microstructure porosity of the concrete [94,95,96], thereby enhancing anti-carbonation resistance.

### 3.6. Carbonation Resistance (Carbonation Coefficient)

According to Valcuende and Parra [97], the concrete resistance towards carbonation is inversely proportional to the carbonation coefficient according to Fick’s first law of diffusion, this relationship can be expressed as follows:(12)$$X=K{\left(t\right)}^{0.5}$$
where K, X and t are the carbonation coefficient (mm/$\sqrt{\mathrm{month}}$), carbonation depth (mm) and exposure time (months), respectively, the results are listed in Table 5. Figure 10 presents the experimental results which display very good correlation factors with the regression lines and clearly demonstrate a linear variation of the carbonation coefficient with the square root of exposure time. The correlation coefficients of modified concrete with GA-NPs have a great value of 0.9093 while that of the control specimen was 0.7877, indicating a strong correlation of the carbonation depth with exposure time in the presence of inhibitors. Moreover, the increasing of the carbonation coefficient in the presence of GA-NPs inhibitor (Table 5) from (3–5.02) compared to that of control specimens (4.1–14.04) after 1 to 3 months of exposure respectively, revealing that the modified specimens exhibited high resistance to carbonation and strengthen the concrete matrix by 64.25%. According to Shaikh and Supit [56], the structure of nanoparticles can assist to affect the improvement of durability properties and strength development. Furthermore, the high surface area of nanoparticles can be consumed the Ca(OH)_2_ in the concrete during the hydration process and development of further C-S-H gel. Moreover, the aggregate structure can be also enhanced in the presence of nanoparticles, resulting in a strong bond between cement paste and aggregates.

### 3.7. Effect of pH Value of Concrete

The pH values of concrete specimens exposed to carbonation for 180 days in the absence and presence of GA-NPs inhibitor at a variety of depths are depicted in Figure 11. In general, the pH values of control specimen are significantly lower than that of concrete treated with green GA-NPs inhibitor. The pH value of the carbonated control specimen dropped to 9.2 at depth of 2 mm which was measured from the cube’s surface and increased slightly to 10.4 at depth of 20 mm. This means that the control concrete specimens have undertaken rapid deterioration, and have been decalcified in a harsh condition, in comparison to the concrete treated with green inhibitor. The accelerated deterioration can be attributed to an extra intense decalcification of control concrete and increasing the penetration of carbon dioxide. The latter reacts with calcium ions in the concrete matrix which leads to consuming more calcium hydroxide and C-S-H, as a result, formation of CaCO_3_. In another meaning, during the carbonation process, the content of Ca^2+^ ions in the pore solution becomes reduced. This, in turn, triggers dissolution of calcium hydroxide and diffusion of Ca^2+^ from the interior of the concrete to the site of carbonation, where the concentration of both components will be at a minimum due to the low solubility of calcium carbonate. Moreover, following long-term degradation, the cementitious materials of the concrete pore to the surface of the specimens was nearly free of calcium. This leads to a reduction in the alkalinity of the pore solution and may lead to decomposition of the hydration products which, in turn, generally leads to increased porosity of cement-based materials. This increased porosity due to leaching, along with the lowered pH values, can enhance the rate of ingress of aggressive substances or the rate of corrosion of reinforcing steel. The chemical reactions of carbonation process may explained as follows [90]:(13)$${\mathrm{CO}}_{2}+H_{2}O\to {\mathrm{HCO}}_{3}^{-}+H^{+}$$
(14)$${\mathrm{HCO}}_{3}^{-}\to {\mathrm{CO}}_{3}^{2-}\left(\mathrm{carbonate}\;\mathrm{ions}\right)+H^{+}$$

In the pore solution, the ${\mathrm{CO}}_{3}^{2-}$ can be reacted with calcium ions as follows,
(15)$${\mathrm{Ca}}^{2+}+{\mathrm{CO}}_{3}^{2-}\to {\mathrm{CaCO}}_{3}$$

As a result of the above reaction, the concentration of calcium ions reduced and dissolution of essentially Ca(OH)_2_.
(16)$${\mathrm{CaOH}}_{2}↔{\mathrm{Ca}}^{2+}+2{\mathrm{HO}}^{-}\left(\mathrm{solubility}\;9.95\times 10^{-4}\right)$$
(17)$${\mathrm{Ca}}^{2+}+{\mathrm{CO}}_{3}^{2-}\to {\mathrm{CaCO}}_{3}\;\left(\mathrm{solubility}\;0.99\times 10^{-8}\right)$$

This leads to dissolved and precipitation of Ca(OH)_2_ and CaCO_3_ respectively until the Ca(OH)_2_ is totally consumed.

Conversely, the carbonation effect on GA-NPs specimens was negligible, since the pH remains as high as 12.5 at a depth of 18 mm (Figure 11) which is comparable to pH in normally hydrated concrete. Obviously, the tendency of the curve is higher in the GA-NPs specimens demonstrating that the effectiveness of nanoparticles on the decreasing of diffusivity. Besides, the high surface area of nanoparticles can be acted as pozzolanic materials, which is enhanced the reaction of concrete compounds [98,99], hence, increased the concrete’s durability by decreasing levels of free calcium hydroxide and developing C-S-H gel. This is reflected in greatly enhanced pH values by up to 18.4% at a depth of 20 mm relative to the control specimen.

### 3.8. Morphologies of Concrete Specimens

#### 3.8.1. Thermogravimetric Analysis (TGA) and Differential Thermal Analysis (DTA)

Figure 12 illustrates the TGA/DTA curves for concrete powder specimens with and without GA-NPS exposed to CO_2_ gas for 180 days. These analyses indicate three characteristic temperature ranges in which weight loss occurs during heating. From Figure 12a,b, the components of powder specimens exhibit a continual weight loss which denotes a change in the mass of the powder. Melar et al. [100] acknowledged that the first region of weight loss of concrete powder appeared at 25 °C to 120 °C is related to the loss of moisture which is mainly absorbed from the atmosphere, in addition to a partial loss of the bound water such as dehydration of the ettringite and interfacial layer (C-S-H). According to Heikal et al. [101] and Kinoshita et al. [102], the second region of weight loss of specimens occurred at a temperature of 400 °C to 550 °C is associated with dehydroxylation of Ca(OH)_2_, while the weight loss of the third region at 600–750 ℃ is assigned to decomposition of amorphous calcium carbonate (CaCO_3_) and crystalline. The dehydrogenation of Ca(OH)_2_ can be briefly described as follow; the amount of the molecules of water are rapidly reduced in the dehydration process, with not much change in the amounts of hydroxyl. However, when the degree of hydration is decreased to half, just a small quantity of water has remained in the C-S-H matrix. Additional reduction of hydration degree can occur by the decreasing of Ca-OH and Si-OH groups “dehydroxylation stage”. Thereafter, a single molecule of H_2_O separates and awards a single of its own H atoms to the deprotonated Silanol, as a result, the development of double hydroxyls, Ca-OH and Si-OH. Moreover, in the presence of deprotonated hydroxyl groups, the H_2_O atoms are thermodynamically unsteady. Hence, the dehydroxylation of calcium–silicate–hydroxide takes place.
(18)$$\mathrm{Si}-\mathrm{OH}+\mathrm{Si}-\mathrm{OH}=\mathrm{Si}-O-\mathrm{Si}+H_{2}O$$
(19)$$\mathrm{Si}-\mathrm{OH}+\mathrm{Ca}-\mathrm{OH}=\mathrm{Si}-O-\mathrm{Ca}+H_{2}O$$
(20)$$\mathrm{Ca}-\mathrm{OH}+\mathrm{Ca}-\mathrm{OH}=\mathrm{Ca}-O-\mathrm{Ca}+H_{2}O$$

Table 6 depicts the amount of CH and C-S-H components of concrete specimens after exposure to CO_2_ for 180 days. The calcium hydroxide content of GA-NPs specimens was decreased significantly from 10.20% to 3.72% in comparison to the specimen without GA-NPs. The presence of the lesser weight-loss values of treated concrete with GA-NPs inhibitor at a temperature range of 400 to 550 ℃ proposed that the consumption of Ca(OH)_2_ (portlandite) resulted from the significant pozzolanic activity of GA-NPs concrete and formation of extra C-S-H gel (9.72%). The improvements of portlandite consumption and formation of C-S-H gel were 63.53% and 67.63% respectively, with respect to control concrete. Nevertheless, it should also be noted that the particularly high amounts of portlandite in control concrete (10.20%) which was determined at the same temperature, can be resulted due to the susceptibility of the concrete to high carbon dioxide gas because of the porous nature of the concrete. Moreover, the modified concrete specimen with GA-NPs has displayed a decrease in the mass loss at a temperature range of 600 to 800 ℃, which is strongly confirmed that the incorporation of green GA-NPs inhibitor made the concrete denser, less porous and less permeable to diffuse the CO_2_ gas, thus, enhanced the durability of concrete.

#### 3.8.2. SEM-EDX

Following 180 days of exposure to accelerated carbonation conditions, the SEM morphologies of the control specimen and the specimen treated with green GA-NPs inhibitor were as presented in Figure 13. Figure 13a, reveals that the hydrated product calcite has formed at the expense of portlandite consumption in the control specimen. Cracks linked to the increase in internal deterioration due to CO_2_ attack are observed on the surface of the control specimen. According to Ekolu [103], diffusion of CO_2_ into the concrete lowers the pH to around 9–10, leading to breakdown of the natural protective passive layer, so that corrosion begins to occur. The CH (calcium hydroxide) and (C-S-H) both react with CO_2_ to form poorly soluble calcite, which precipitates within the pore space, affecting the properties of the concrete and increasing the corrosion rate of the embedded steel.

In contrast, Figure 13b indicates that the incorporation of 3% GA-NPs inhibitor resulted in total improvement of the concrete surface morphology, revealing a smooth concrete surface with the complete absence of calcite or cracks. According to Mukharjee and Barai [104], the nanoparticles can be reacted rapidly with crystalline (CH) to form C-S-H, i.e., the crystals can be absorbed. This decreases the size and quantity of CH crystals while the C-S-H gel makes up about 70% of the hydration products and contributes towards filling the voids and so enhancing the binding paste matrix and the density of the interfacial transition zone (ITZ). The nanoparticles are able to fill the remaining voids in the C-S-H gel structure, thus further enhancing the density of the binding paste matrix. Furthermore, the nanoparticles create a strong bond with C-S-H gel particles via acting as a nucleus in the C-S-H gel structure.

Figure 14 presents the EDX spectra of the concrete specimens, while Table 7 lists the percentage atomic content of various elements in the concrete surface, as determined by EDX. The penetration of CO_2_ gas into the concrete surface reduces both the pH value and the concentration of Ca^2+^ ions, leading to dissolution of the calcium–silicate–hydroxide gel due to consumption of CH. Decomposition of monosulfate and ettringite occur respectively at pH 11.6 and 10.6, after which most of the Ca^2+^ ions from the C-S-H gel bind to CaCO_3_ leaving just a minor amount of Ca^2+^ ions in the silica gel. Hussain et al. [105] attribute this to the reduction in the quantity of portlandite by carbonation, resulting in low Ca/Si ratios in the C-S-H gel, and Pelisser et al. [106] pointed to an increase in concrete durability when the Ca/Si molar ratio of the C-S-H gel fell below 1.65. Examining Table 7 and Figure 14a in the light of this hypothesis, the Ca/Si ratios of control specimens (3.22%) are seen to be greater than the normal acceptable value, thus explaining the observed pore structures. By contrast, the EDX spectrum presented in Figure 14b confirms that the Ca/Si ratio of 0.69% obtained in the presence of GA-NPs inhibitor is due to the incorporation of alkali metal into the C-S-H gel.

#### 3.8.3. X-ray Diffraction (XRD)

The X-ray powder diffraction (XRD) results for the control concrete specimen and the specimens treated with green GA-NPs inhibitor, following 180-day of exposure to CO_2_ gas, are presented in Figure 15. Each concrete specimen showed peaks at 2θ° = 20.9°, 26.9°, 39.8°, 50.5° and 60.2° due to the presence of quartz (Q), at 2θ° = 36.1°, 54.4° and 63.6° due to portlandite (P) and at 2θ° = 29.7°, 40.3°, 67.7°, 73.4° and 75.6° due to calcite (C). Nevertheless, the specimen treated with GA-NPs inhibitor showed reduced peaks for portlandite and calcite along with increased peaks for quartz in comparison with the control specimen under the same curing conditions. This suggests that the incorporation of the GA-NPs inhibitor could influence both the degree of portlandite and calcite crystallisation and the crystal sizes by filling and obstructing the pore structure in the concrete surface layer. The resulting reduction in connectivity and pore size would reduce the permeability of the concrete to diffusion of CO_2_.

## 4. Conclusions

The impact of the developed green gum Arabic inhibitor upon the behaviour of reinforced concrete exposed to carbonation for 180 days was investigated. The conclusions are set out according to the results of various tests as follows:The results of weight loss and EIS evidenced that the incorporation the GA-NPs inhibitor into concrete increased the inhibition efficacy and decreased the corrosion rate up to 94.5% and 0.57 × 10^−3^ mm/year respectively, by increasing the surface coverage area led to increasing the adsorption of inhibitor molecules on steel reinforcement. In addition, the results of EIS revealed that the double-layer capacitance of steel reinforcement was decreased while the charge transfer resistance was increased in the presence of inhibitor due to the formation of a protective layer.The morphology of steel reinforcement by SEM and AFM studies confirmed the formation of a protective layer. In addition, AFM images of rebar in the presence of GA-NPs inhibitor confirmed low surface roughness which is strongly promoted the ability of the inhibitor to adsorb over rebar.The carbonation depth tends to increase with increased time of accelerated curing. Nevertheless, the lowest level of carbonation was clearly observed for the specimen modified with GA-NPs, indicating that the tiny scale of green inhibitor molecules, which was 40.24 nm according to TEM, can block the capillary pores of concrete to form an impermeable barrier against the ingress of CO_2_.The best linear relationship was observed for the concrete modified with 3% GA-NPs inhibitor, with a correlation coefficient of 0.9093, compared to a minimum value of 0.7877 for the control specimen.The alkalinity of control specimens was reduced after 180 days of exposure to CO_2_ gas by dropping the pH value to 10.4 at depth of 20 mm. In contrast, the high surface area of nanoparticles was capable to fill the porous of concrete, which led to enhancing the pozzolanic reaction and formation of extra C-S-H gel, hence, the pH value of modified concrete was raised to 12.5 at a depth of 20 mm.Finally, the microstructural morphologies of concrete specimens in the presence of GA-NPs inhibitor including TGA/DTA, SEM-EDX and XRD confirmed the ability of green GA-NPs inhibitor to increase the durability of concrete by reducing the ratio of Ca/Si to 0.69, consumption of portlandite, and increased the peaks of quartz, thus the development of extra C-S-H gel.

## Figures and Tables

**Figure 1 materials-14-07867-f001:**
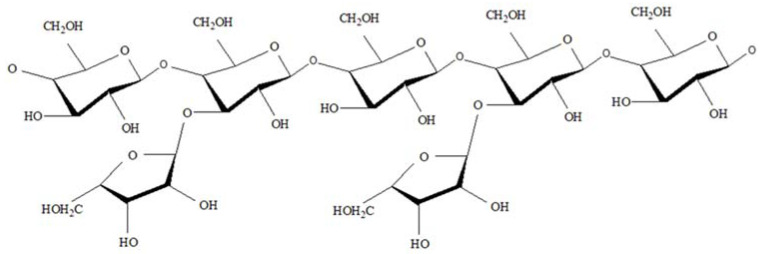
The chemical structure of arabinogalactan [67].

**Figure 2 materials-14-07867-f002:**
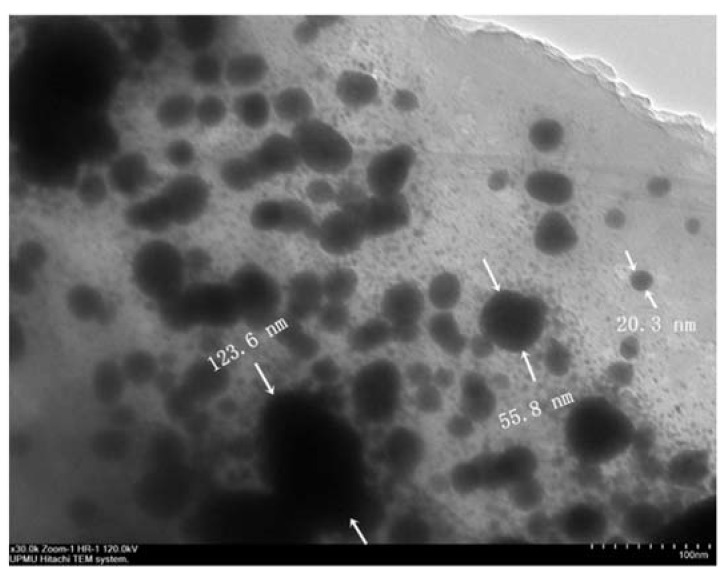
TEM image of GA-NPs inhibitor.

**Figure 3 materials-14-07867-f003:**
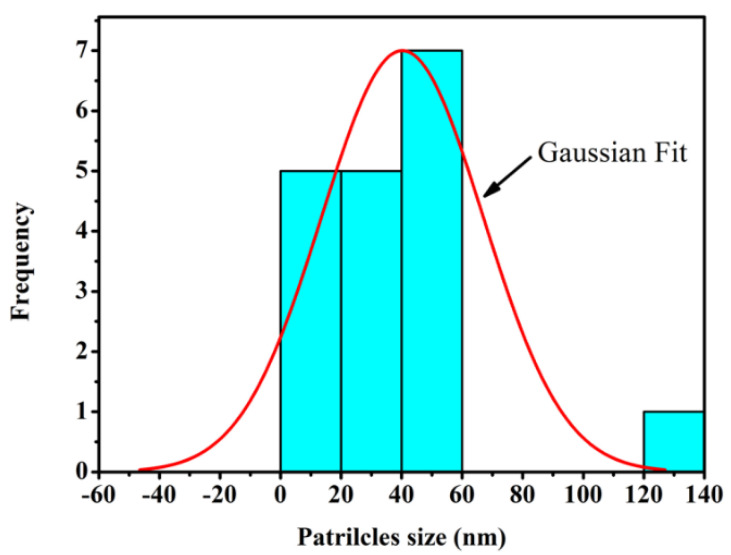
Distribution of GA-NPs according to the TEM image in sync with Gaussian fitted curve.

**Figure 4 materials-14-07867-f004:**
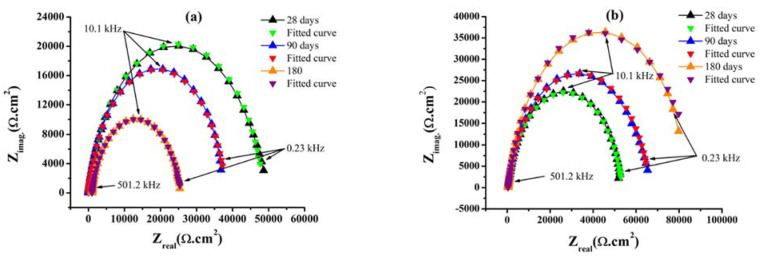
Nyquist plots of steel rebars embedded in concrete exposed to CO_2_, (**a**) control and (**b**) treated concrete with green GA-NPs inhibitor.

**Figure 5 materials-14-07867-f005:**
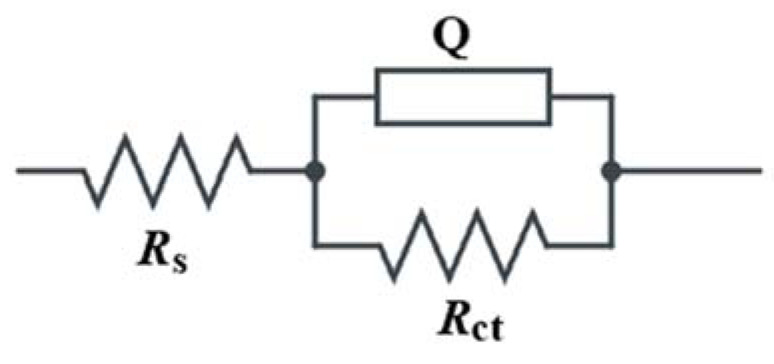
Equivalent circuit employed for fitting the impedance results.

**Figure 6 materials-14-07867-f006:**
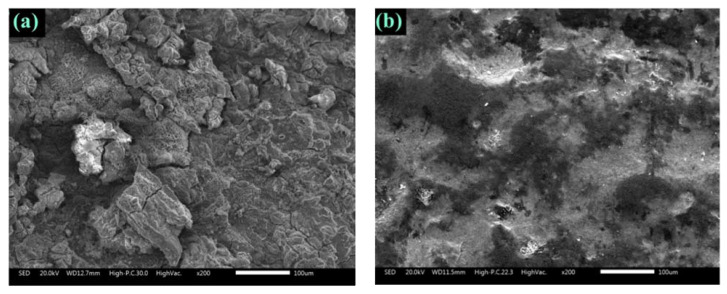
Surface morphology of steel reinforcement after 180 days of exposure to CO_2_, (**a**) control and (**b**) modified concrete with GA-NPs.

**Figure 7 materials-14-07867-f007:**
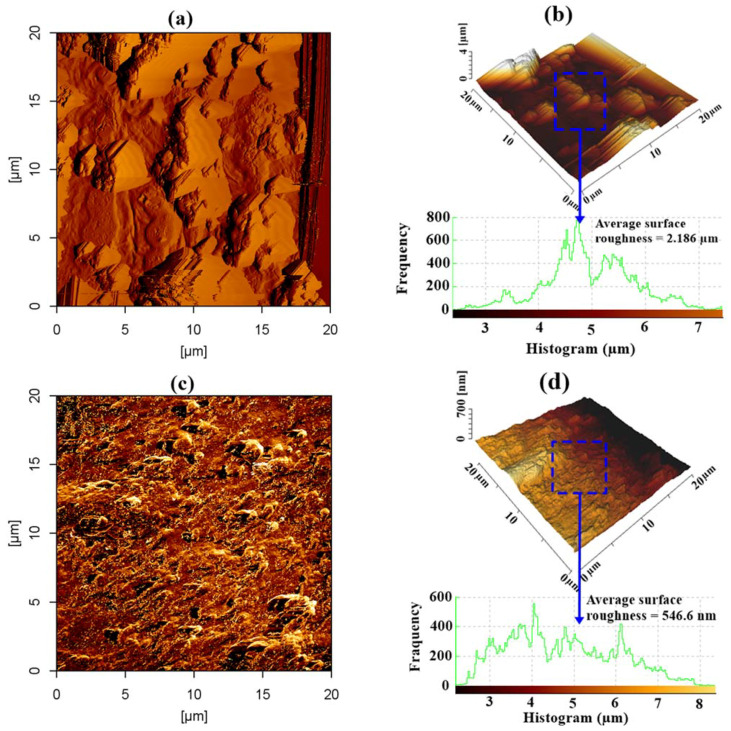
The AFM micrograph for steel reinforcement surfaces exposed to CO_2_ for 180 days, (**a**,**b**) 2D, 3D images for control and (**c**,**d**) 2D, 3D images for treated specimens with GA-NPs.

**Figure 8 materials-14-07867-f008:**
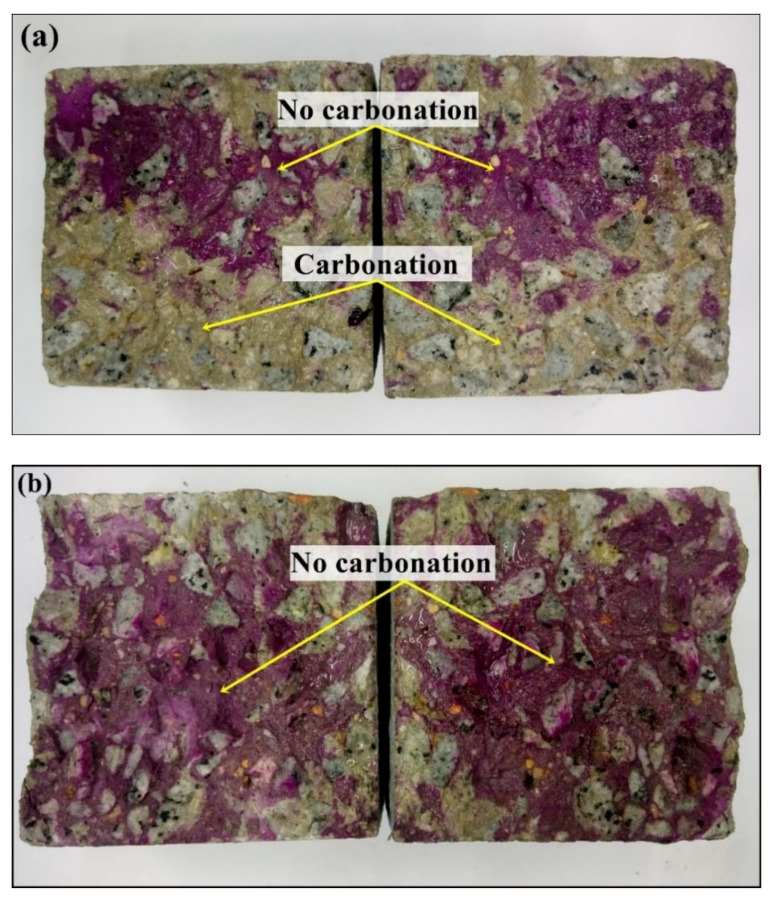
Accelerated carbonation for concrete specimens (**a**) Control, (**b**) GA-NPs inhibitor.

**Figure 9 materials-14-07867-f009:**
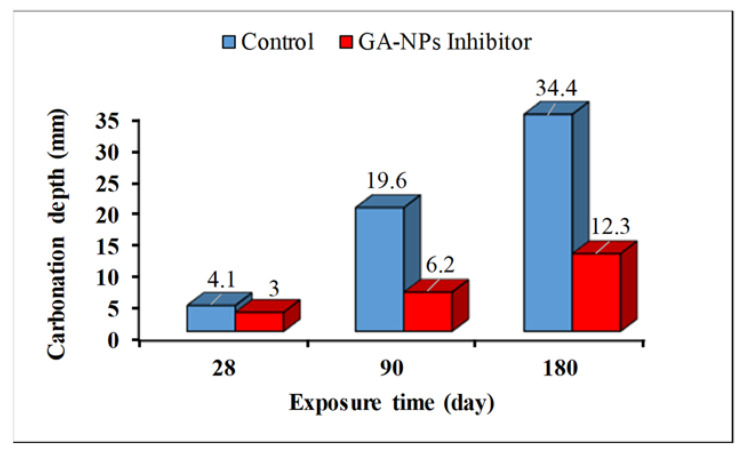
Carbonation depth of concrete specimens at various exposure time.

**Figure 10 materials-14-07867-f010:**
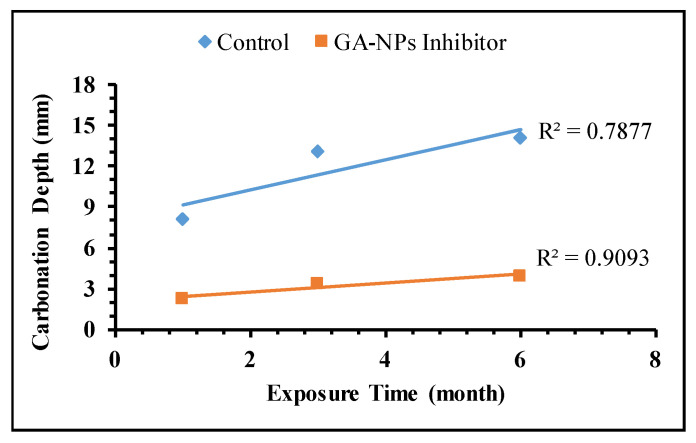
Carbonation coefficient of concrete specimens.

**Figure 11 materials-14-07867-f011:**
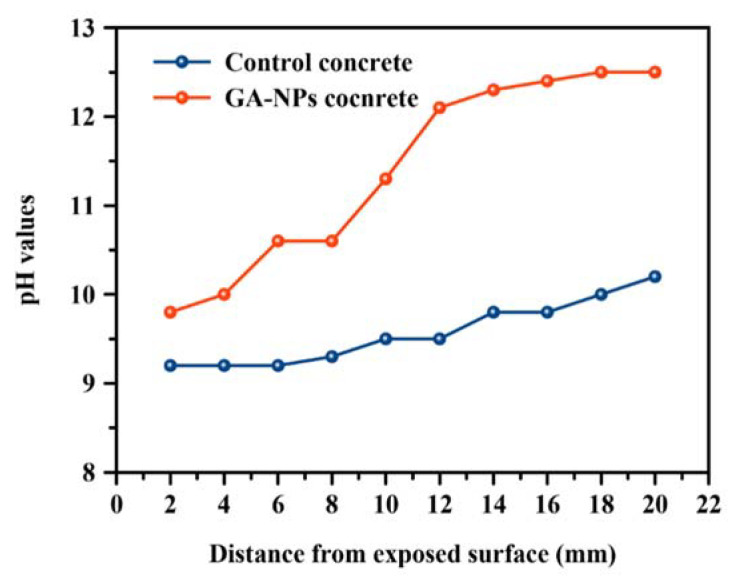
pH values of carbonated concrete after 180 days of exposure to CO_2_.

**Figure 12 materials-14-07867-f012:**
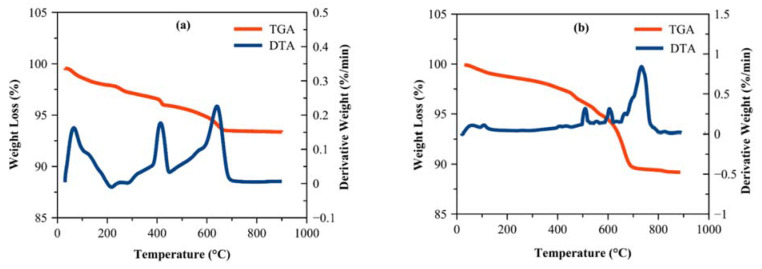
TGA/DTA analysis for concrete specimen exposed to CO_2_ gas for 180 days (**a**) Control, (**b**) GA-NPs.

**Figure 13 materials-14-07867-f013:**
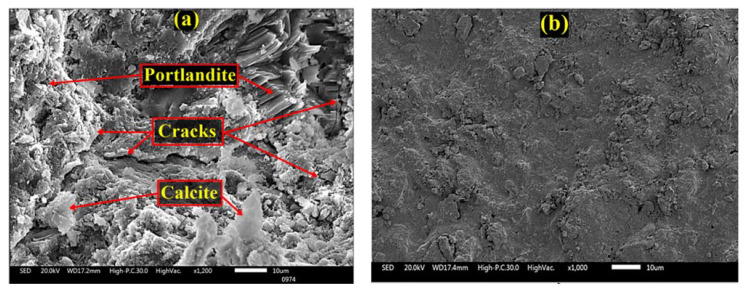
Morphology of concrete specimens exposed to carbon dioxide gas (**a**) control, (**b**) GA-NPs concrete.

**Figure 14 materials-14-07867-f014:**
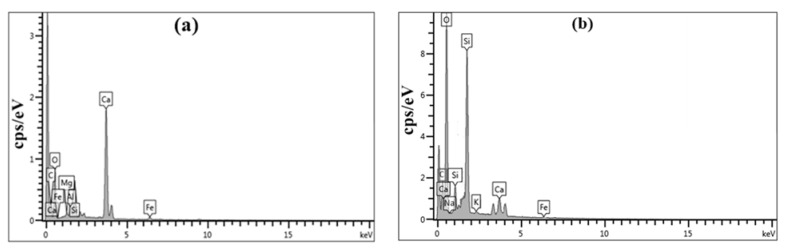
EDX spectra of concrete specimens exposed to carbon dioxide gas (**a**) control, (**b**) GA-NPs concrete.

**Figure 15 materials-14-07867-f015:**
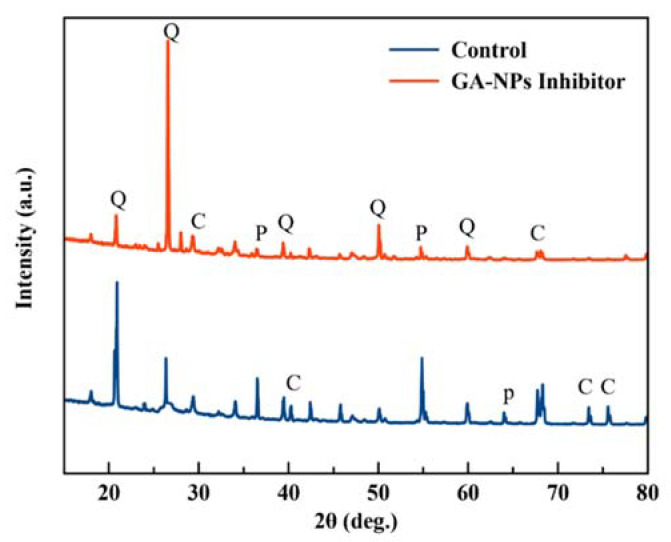
XRD patterns of concrete specimens exposed to CO_2_ gas for 180 days.

**Table 1 materials-14-07867-t001:** The chemical composition of *Acacia Senegal* (gum Arabic).

Element	g/kg
Crude fibre	73.2–79.8
Protein	25.0–37.1
Dry matter	870–877.9
Nitrogen free extract	851.5–868.1
Ether extract	1.8–4.3
Ash	24.7–27.3
pH	4.69
Galactose	389
Arabinose	257
Glucuronic acid	215
Rhamnose	95

**Table 2 materials-14-07867-t002:** Concrete mix design per cubic metre.

Item	Cement (kg/m^3^)	Aggregate (kg/m^3^)		w/c (kg/m^3^)	Inhibitor (kg/m^3^)
		Coarse	Fine		
Control concrete	395	682	1106	217	-
Green GA concrete	395	682	1106	217	11.85

**Table 3 materials-14-07867-t003:** Weight loss parameters for steel reinforcement in the absence and presence of green GA-NPs inhibitor.

Weight Loss Parameters					
Time (day)	Specimen	Weight Loss (g)	CR × 10^−3^ (mm/year)	IE (%)	Surface Coverage (θ)
28	Control	0.0953	5.63	-	-
	GA-NPs	0.0171	1.02	81.95	0.820
90	Control	0.2363	6.98	-	-
	GA-NPs	0.0315	0.93	86.67	0.867
180	Control	0.6775	9.87	-	-
	GA-NPs	0.0393	0.57	94.20	0.942

**Table 4 materials-14-07867-t004:** EIS data for reinforcing steel-concrete slabs in the absence and presence of 3% GA-NPs exposed to CO_2_ environment.

Exposure Time (Day)	Specimen	EIS Data				
		$R_{s}$ (Ω cm^2^)	$R_{\mathrm{ct}}$ (Ω cm^2^) × 10^2^	$C_{\mathrm{dl}}$ (μF cm^−2^) × 10^−5^	IE (%)	θ
28	Control	10.8	35.06	3.06	-	-
	GA-NPs	11.6	204.00	0.0243	82.8	0.83
90	Control	11.1	33.84	3.19	-	-
	GA-NPs	11.9	276.70	0.0213	87.8	0.88
180	Control	11.7	27.12	7.68	-	-
	GA-NPs	12.8	496.00	0.0138	94.5	0.95

**Table 5 materials-14-07867-t005:** Parameters of carbonation resistance.

Specimens	t (month)	X (mm)	$K\;(\mathrm{mm}/\sqrt{\mathrm{month}})$
Control concrete	1	4.1	4.1
	3	19.6	5.54
	6	34.4	14.04
GA-NPs concrete	1	3.0	3
	3	6.2	3.58
	6	12.3	5.02

**Table 6 materials-14-07867-t006:** Amount of CH and C-S-H gel in the concrete matrix after exposure to CO_2_ environment for 180 days.

Specimen	CH (%)	C-S-H (%)
Control concrete	10.20	3.00
GA-NPs concrete	3.72	9.27

**Table 7 materials-14-07867-t007:** The percentage atomic content of various elements in the concrete surface exposed to CO_2_ gas for 180 days.

Specimen	Components (%)									
	O	Ca	C	Si	Al	Mg	Fe	Na	K	Ca/Si %
Control	46.7	25.1	13.3	7.8	2.9	1.1	2.3	0.8	0.0	3.22
GA-NPs	49.5	11.6	17.2	16.7	1.8	0.0	1.5	0.0	1.7	0.69

## Data Availability

Data sharing not applicable.

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
