# Peer review of "Gum Arabic Nanoparticles as Green Corrosion Inhibitor for Reinforced Concrete Exposed to Carbon Dioxide Environment"

_materials, 2021, doi:10.3390/ma14247867_

Round 1
Reviewer 1 Report
In this article, the following deficiencies must be eliminated.
1- Please modify the title of the article.
2- Include more references on nanoparticle publications in the introduction part of the article. The ones I could find are as follows.
3- Please talk about the natural material you use and its technical features.
*Guler, S., TürkmenoÄŸlu, Z. F., & Ashour, A. (2020). Performance of single and hybrid nanoparticles added concrete at ambient and elevated temperatures. Construction and Building Materials, 250, 118847. doi:10.1016/j.conbuildmat.2020.
3- Please write the results part of the study by summarizing the results you have obtained a little more.
Author Response
Your invaluable comments highly appreciated, please see the attached file.

Reviewer 2 Report
Please find the attached file.

Author Response
Your invaluable comments to enhance the manuscript greatly appreciated.

Round 2
Reviewer 2 Report
Well done.